# BioChemDDI: Predicting Drug–Drug Interactions by Fusing Biochemical and Structural Information through a Self-Attention Mechanism

**DOI:** 10.3390/biology11050758

**Published:** 2022-05-16

**Authors:** Zhong-Hao Ren, Chang-Qing Yu, Li-Ping Li, Zhu-Hong You, Jie Pan, Yong-Jian Guan, Lu-Xiang Guo

**Affiliations:** 1School of Information Engineering, Xijing University, Xi’an 710123, China; zhonghaoren98@gmail.com (Z.-H.R.); gyjfs926@163.com (Y.-J.G.); guoluxiang08@gmail.com (L.-X.G.); jiepan960930@gmail.com (J.P.); 2College of Grassland and Environment Sciences, Xinjiang Agricultural University, Urumqi 830052, China; 3School of Computer Science, Northwestern Polytechnical University, Xi’an 710129, China; zhuhongyou@nwpu.edu.cn

**Keywords:** drug–drug interactions, attention mechanism, natural language processing, multi-level information and graph collapse

## Abstract

**Simple Summary:**

Throughout history, combining drugs has been a common method in the fight against complex diseases. However, potential drug–drug interactions could give rise to unknown toxicity issues, which requires the urgent proposal of efficient methods to identify potential interactions.We use computer technology and machine learning techniques to propose a novel computational framework to calculate scores of drug–drug interaction probability for simplifying the screening process. Additionally, we built an online prescreening tool for biological researchers to further verify possible interactions in the fields of biomedicine and pharmacology. Overall, our study can provide new insights and approaches for rapidly identifying potential drug–drug interactions.

**Abstract:**

During the development of drug and clinical applications, due to the co-administration of different drugs that have a high risk of interfering with each other’s mechanisms of action, correctly identifying potential drug–drug interactions (DDIs) is important to avoid a reduction in drug therapeutic activities and serious injuries to the organism. Therefore, to explore potential DDIs, we develop a computational method of integrating multi-level information. Firstly, the information of chemical sequence is fully captured by the Natural Language Processing (NLP) algorithm, and multiple biological function similarity information is fused by Similarity Network Fusion (SNF). Secondly, we extract deep network structure information through Hierarchical Representation Learning for Networks (HARP). Then, a highly representative comprehensive feature descriptor is constructed through the self-attention module that efficiently integrates biochemical and network features. Finally, a deep neural network (DNN) is employed to generate the prediction results. Contrasted with the previous supervision model, BioChemDDI innovatively introduced graph collapse for extracting a network structure and utilized the biochemical information during the pre-training process. The prediction results of the benchmark dataset indicate that BioChemDDI outperforms other existing models. Moreover, the case studies related to three cancer diseases, including breast cancer, hepatocellular carcinoma and malignancies, were analyzed using BioChemDDI. As a result, 24, 18 and 20 out of the top 30 predicted cancer-related drugs were confirmed by the databases. These experimental results demonstrate that BioChemDDI is a useful model to predict DDIs and can provide reliable candidates for biological experiments. The web server of BioChemDDI predictor is freely available to conduct further studies.

## 1. Introduction

Drugs play a crucial role in curing diseases and enhancing quality of life [1]. During drug development, drug–drug interactions (DDIs) are a critical consideration and the drug targeting of the selected protein should be bioavailable (e.g., favorable absorption and metabolism) [2]. However, potential DDIs may lead to a strong rise or drop in the plasma concentration of the drug or metabolite, and even generate toxic compounds [3]. From a clinical perspective, a combination of drugs is used for the treatment of complex diseases, but unexpected DDIs may induce adverse reactions, which can give rise to drug withdrawal and even to the death of the patient [4,5]. Thus, the early identification of potential DDIs is very critical for drug development and medical safety.

Alterations in drug pharmacokinetics and drug pharmacodynamics can be caused by DDIs [6]. Pharmacokinetic DDIs crop up when the perpetrator drug disrupts the absorption, distribution, metabolism and elimination (ADME) of the victim drug, and also when the perpetrator drug interacts with the protein of the victim drug or other protein within the same signaling pathway [4]. To screen and analyze unknown DDIs, biological techniques are mainly used, which are regarded as the ultimate way to judge and validate the DDIs, containing metabolism-based and transporter-based DDIs, such as testing whether the drug is the inhibitor/inducer or substrate of CYP enzymes [7], and testing whether the drug is the inhibitor/inducer or substrate of a P-gp transporter [3]. Then, based on the in vitro parameters, a cumbersome dynamic model (e.g., PBPK) and expensive in vivo experiment should be constructed and analyzed for the final validation.

Traditional biological technologies face the challenges of high cost, limited participant number, low efficiency and large number of pairwise drugs waiting for identification [2]. Additionally, the constantly increasing demand for drug therapy makes the identification of potential DDIs before clinical medications are administered [8] more and more urgent. Consequently, exploiting large-scale computational prediction methods as a decision aids for a large number of DDI candidates prescreening to provide a direction or prioritize of the in vitro–in vivo experiments, and improve efficiency for DDI research and development (R&D). Computational methods have gained concern from the academy and the industry, due to their promise to discover drug–drug interactions on the large scale [9,10]. The performance of computers has been greatly improved as the precondition to consider computational methods to realize DDI prediction. Furthermore, researchers constructed many reliable bioinformatic databases on drugs through many biomedical experiments, such as DrugBank [11], ChEBI [12], PubChem [13] and KEGG [14]. Recently, several computational methods have been proposed to reduce the cost of predicting potential DDIs. These methods can be roughly divided into four categories: similarity-based, network-based, matrix-factorization-based and semanticity-based.

Similarity-based methods are one of the relatively main approaches, assuming that, if drugs have similar function structures, they are more likely to have a similar interaction structure. Gottlieb et al. put forward a model named INDI, extracting feature vectors by calculating seven drug similarities and predicting interactions of the drugs by logistic regression [15]. Cheng et al. merged many drug similarities to express drug–drug pairs and exploited five classifiers to build predicting models [16]. Ferdousi et al. provided a method to construct embedding vectors of drugs, using four biological elements, including carriers, transporters, enzymes and targets (CTET), to predict potential DDIs through a Russell–Rao similarity [17]. Rohani et al. predicted DDIs based on fusing similarity matrices and tested the performance on three different scales of drug similarity datasets [18]. Due to the structure information of the interaction network and chemical sequence information not being considered, the final prediction effect was not very good.

Network-based methods infer novel drug–drug edges by the topological structure of the network and biological network that involves biomedical entities, or learning the high-order drug similarity and the propagating similarity. Zhang et al. predicted DDIs using an integrative label propagation method with high-order similarity transitivity on the multi-scale similarity-based network. It also can rank multiple drug information sources [19]. Park et al. exploited the random walk with restarting on the protein–protein network method to the analog propagation of signals to predict drug–drug interactions [20]. Deepika et al. proposed a meta-learning framework, extracting representation on four types of feature networks through Node2vec and containing chemical feature networks, biological feature networks, phenotypic feature networks and disease feature networks, then integrating the results of four classifiers to predict unknown DDIs [21]. Liu et al. drew an approach DANN-DDI that contains five types of drug networks and learned the features through the graph embedding algorithm SDNE. An attention neural network is designed to learn concatenating representation and a deep neural network is used to generate prediction results [22]. Although these methods have shown a good prediction performance, most of them preserve higher-order structure features with difficulty and stick to the local optimum with ease. The attention mechanism is not used to process multiple pieces of information, which leads to a limited performance.

Matrix-factorization-based methods turn the DDI adjacency matrix into several decomposed matrices and then re-establish adjacency by the decompositions. The model, named TMFUF, was developed by Shi et al. to identify DDIs, which uses drug additional information to rebuild the interaction matrix through triple matrix factorization [23]. Zhang et al. proposed an ensemble model that is based on sparse feature learning for predicting drug–drug interactions [24]. Zhang et al. designed a method for DDIs prediction that uses eight types of background information based on the matrix factorization of a manifold regularization [25]. Yu et al. proposed DDINMF applying semi-nonnegative matrix factorization to conclude the enhanced and degressive prediction of pairwise drugs [26]. Shi et al. introduced a BRSNMF model, an optimization of the DDINMF model, technically utilizing drug-binding protein as the feature to predict DDIs of new drugs [27]. These approaches exploit the biological information of many supplements to ensure generalization, but the important information on the chemical sequence is not fully considered.

Semanticity-based methods generally abstract the information from the semanticity of sentences about drugs through text-mining, and then the candidates of drug interactions are detected and classified. Chowdhury et al. applied a framework that can extract information on multi-phase relations, exploiting the scope of negation cues and semantic roles to reduce the skewness of the data, and used SVM to calculate the possibility of DDIs [28]. Zhu et al. exploited the BioBERT method to pre-train word vectors of drug descriptions and extracted the semantic representation of sentences by the BiGRU, integrating drug entity information by entity-aware attention, obtaining prediction results by the MLP [29]. However, these methods are heavily dependent on the clinical evidence in the post-market, which means that there is no capability of providing potential DDI alerts before clinical medications are administered.

Although the aforementioned methods have their own advantages and play a crucial role in computational method development for drug–drug interaction prediction, there are still some limitations. (i) Due to several existing relevant computational methods only focusing on single information of drugs, they still cannot satisfy the demand for prediction accuracy in reality applications. (ii) Furthermore, most of them rely on artificially designed molecular representation, limited by the knowledge of domain experts. (iii) Ignoring deep network structure information and integrating drug features without the attention machine could lead to limited prediction performance.

In in silico research, data availability and accessibility are crucial factors in determining the accuracy and precision of calculational methods [30]. In this paper, in order to address the existing deficiencies, we propose a novel framework for fusing drug chemical sequence, drug biological function similarity and deep network topology structure with an attention machine (BioChemDDI) to predict potential DDIs. In particular, we first obtain the drug biochemical features regarding chemical sequence information and biological function information through a word-embedding algorithm and the similarity matrix fusing method, respectively. Notably, the chemical sequence feature of each drug is firstly represented as a matrix, whose dimension can be reduced by a Convolutional Neural Network (CNN). Then, the rich structural information of the interaction network is extracted by efficiently graph embedding with graph collapse. Thirdly, we introduce the attention mechanism to fuse multiple features and enhance the feature of each drug node. Finally, the interaction scores are generated by the fully connected layer. The proposed model is trained end-to-end, and each feature vector can be further screened for the characteristic factors through the hidden layers. Additionally, in this work, four datasets are used to verify the robustness of our model and are compared with state-of-the-art methods to demonstrate the high efficiency of BioChemDDI. Then, the results of the five-fold cross-validation and the case studies further investigate whether our model is suitable for interaction studies between drugs. More meaningfully, case studies related to three cancer diseases can give insight to reveal unknown drug–drug interactions for inferring combinations of the drug in treating complex diseases. The BioChemDDI model can provide accurate predictions for the potential DDIs and is anticipated to serve as the prioritization tool for the development of drug and clinical applications, which can be used as pre-screening tools for potential DDIs. The computational platform web server is accessible at: http://120.77.11.78/BioChemDDI/ (accessed on 11 April 2022).

## 2. Materials and Methods

### 2.1. Dataset Description

In this study, to demonstrate the robustness of BioChemDDI, we used four datasets called DS1, DS2, D-DS3 and E-DS3. The DS1 was processed by Ren et al. [31], and its biochemical information was downloaded from DrugBank [11], containing one chemical sequence information and four types of biological function information, which are expressed by drug-related receptor-types of carriers, enzymes, targets and transporters. It also includes 1940 drug nodes, and the 219,247 known pairwise DDIs have been proved [32]. Zhang et al. [33] collected DS2 from the databases of DrugBank [11], SIDER [34], KEGG [35], PubChem [13] and OFFSIDES [36]. Additionally, the D-DS3 and the E-DS3 were processed by Shi et al. [27], which, respectively, contain 1562 drugs with 55,278 depressive interactions and 125,298 enhancive interactions and 4 types of biological functions, the tsame as the DS1. Table 1 illustrates his information in more detail.

Noteworthily, due to the use of testing datasets from different sources, high unbalanced data regarding biological function information is contained, which can be addressed by the biological function fusing module. Known drug–drug pairs were considered positive samples and the negative samples can be presented by other drug–drug pairs in the network. To avoid bias from the imbalance of positive and negative samples, the negative samples were randomly chosen to be equivalent to the positive samples. We used a 5-fold cross-validation and split train–test on our datasets to proceed with a meaningful contrast. After disrupting the samples, 70% of them were seen as training datasets. We treated 20% of them as test datasets, and regarded the other 10% as validation datasets. Inspired by Feng et al. [37], different scales of datasets were used to validate the superior performance of BioChemDDI. Because DS2 is often utilized to evaluate the ability of prediction, we also used DS2 as the primary comparison experimental dataset to save time and costs. DS1, D-DS3 and E-DS3 were selected to evaluate the stability of BioChemDDI.

### 2.2. Overview of the Methods

In this paper, we propose a learning framework, BioChemDDI, to efficiently predict potential DDIs. As Figure 1 illustrates, BioChemDDI consists of four modules comprising a drug chemical sequence information learning module, drug–drug network structural information learning module, biological function information learning module and attentional multi-feature integration module. At the beginning, the chemical sequence was indicated by learning their corresponding SMILES through the NLP method of CBOW. Then, we represented each type of biological function as the adjacency matrix and calculated the drug similarity matrix under different function types. After that, all of the similarity matrixes were fused efficiently by SNF. Before Hierarchical Representation Learning for Networks (HARP) [38] was used to extract drug network structural information, we constructed the DDI network from the positive samples in the train set. Furthermore, the attentional integration mechanism integrates biochemical features and network structural features to finally predict potential DDIs. Additionally, we used the binary cross-entropy loss and error backpropagation (BP) feedforward network to implement end-to-end training. A detailed description is given below, according to the order of the function module in the flowchart.

### 2.3. Drug Chemical Sequence Information

With deep learning technology springing up, the methods of word embedding are utilized to raise downstream task performance. In order to make full use of the chemical sequence of drugs, researchers regard SMILES as sentences and treat each symbol of a chemical element or atom as a word. Chemical sequence learning consists in transforming each symbol into a digital vector, thereby each drug sequence can be represented as a matrix. Based on the “natural biological language”, chemical sequence information is learned by word2vec [39], which is a method for learning distributed vector representation of words through a large corpus. CBOW is a word2vec model, which calculates the probability of a certain center word according to the context of the word. It contains an input layer, a hidden layer and an output layer. The embedding vectors of each context word are calculated by the weight matrix ω. Additionally, the arithmetic average of embedding vectors h of the center word is calculated by the hidden layer, as follows:(1)$$h=\frac{1}{c}\omega^{⊤}(x_{1}+x_{2}+\ldots +x_{c})$$
where $x_{C}$ is the one-hot vector representation of *C*-th context words, which are the *V* dimension. The *h* is the average of the embedding vectors of context words. By the weight matrix $\omega^{\prime }$ of the output layer, the co-occurrence probabilities are calculated, and each word is the center word to calculate the embedding vector. Moreover, the probability of the actual central word occurring is maximized by the function:(2)$$E=-\log P(\omega_{O}|\omega_{I1},\omega_{I2},\ldots ,\omega_{IC})=\log\sum_{j^{\prime }=1}^{V}\exp({v^{\prime }}_{\omega j}{}^{⊤}\times h)-{v^{\prime }}_{\omega o}{}^{⊤}\times h$$
where ${v^{\prime }}_{\omega j}$ means the *j*-th row of the weight matrix $\omega^{\prime }$. In this paper, SMILES was seen as the drug structure corpus to make use of the word embedding method. The Python package of gensim was used to obtain the drug feature matrix. Then, in addition to the parameters of “size” being set to 64 and “min_count” being set to 1, the parameter values were default. So, as shown in Figure 2, each symbol is shown as a 64-dimension vector, thereby obtaining the drug feature matrix.

### 2.4. Network Structural Information

The network structural information is a crucial character for potential interaction analysis and prediction. Under the entire drug network, each node can be represented by the structural relationship of nodes. In this subsection, we focus on the task of extracting the embedding vector-containing node structural information. The node structure is more similar, and the embedding vectors of the nodes are closer to each other. To fully utilize the network relation of DDIs and to globally express the straight or potential information flow between nodes, we formulated graph embedding representation by a method named HARP, proposed by Chen et al. [38]. Recently, for large-scale networks, some existing learning algorithms of network representation demand complex computational complexity and many of them are local embedding methods. Relationships of long-distance global network may be ignored by focusing on local embedding configurations, and the learning representations are incapable of revealing vital global patterns of distribution [38]. HARP-LINE, improved from LINE, is suitable for embedding a large global information network into a low-dimensional space of vectors and can catch the deep structure of the interaction network. HARP-LINE consists of three parts: graph coarsening, graph embedding and representation refining, as Figure 3 shows. For a given graph G=(V,E), finding a mapping function $\Phi :V\to R^{|V|\times d},d\ll |V|$ is the first important task of graph representation.

In order to gain a better global embedding, HARP firstly performs a graph coarsening operation, and then a small subgraph $G_{S}=(V_{S},E_{S})$ is generated, in which $\left|V_{S}|\ll |V\right|$ and $\left|E_{S}|\ll |E\right|$. In this way, because of the smaller subgraph containing more coarse-grained information, embedding learning on $G_{S}$ can gain a wider global structure, although the learning method is based on a partial structure. Therefore, as the original graph structure collapses constantly until the number of nodes reaches a certain threshold, LINE [40] is exploited to learn representations in the part of graph embedding. LINE defines the first-order proximity and second-order proximity to indicate, respectively, the nodes’ direct relationship and the relationship of nodes that are not direct, which have a common neighbor node. If the nodes share the same neighbor, they are proximal to each other. The co-occurrence probability of vertexes, $v_{j}$ and $v_{i}$, keeping a similar context can be defined as:(3)$$p_{2}(v_{j}|v_{i})=\frac{\exp(u_{j^{\prime }}{}^{⊤}\times u_{i})}{\sum_{k=1}^{\left|V\right|}\exp(u_{k^{\prime }}{}^{⊤}\times u_{i})}$$
where $u_{i}$ and ${u_{j}}^{\prime }$ are, respectively, the vector representation of the vertex of $v_{i}$ and its context $v_{j}$, and |V| indicates the quantity of context vertex. To assure proximity, the following objective function should be minimized:(4)$$minimizeO_{2}=-\sum_{i\in V}\lambda_{i}d(\bar{p_{2}}(\cdot |v_{i}),p_{2}(\cdot |v_{j}))$$

The function of d(⋅,⋅) is used to calculate the distance of distribution, $\lambda_{i}$, viewed as the importance of $v_{i}$; the empirical distribution probability $\bar{p_{2}}(\cdot |v_{i})$ is designed as:(5)$$\bar{p_{2}}(v_{j}|v_{i})=\frac{w_{ij}}{degree_{i}}$$
where $w_{ij}$ is the weight between $v_{j}$ and $v_{i}$, and $degree_{i}$ stands for the degree of the vertex $v_{i}$. KL divergence is utilized to measure the similarity of distributions. For the sake of simplicity, $\lambda_{i}$ was set in this paper as the degree of $v_{i}$, i.e., $\lambda_{i}=degree_{i}$ and certain constants were left out. The loss function can be reduced as follows:(6)$$minimizeO_{2}=-\sum_{(i,j)\in V}w_{ij}\log(p_{2}(v_{j}|v_{i}))$$

After obtaining the subgraph nodes embedding vectors, using them to initialize the node of the upper-level subgraph can finely collect global information of the coarse graph. Finally, improved embedding vectors can be obtained after optimizing coarse graph embedding by upward iteration. We used the default parameters of the HARP Python package. As all graph embedding methods have a common drawback, that is, that extremely few nodes of the test set do not exist in the train set, these nodes cannot be embedded by graph embedding methods. The biochemical features can increase the information content of the feature to some extent and reduce the impact of the missing graph structural information on the model performance. On the one hand, in reality, many new drugs may have no interaction with known drugs, which can be regarded as no embedded nodes in the network. On the other hand, a few drugs missing some information can also indicate the generalization of our model. In this work, the no embedded nodes were given priority to use the embedding characteristics of their nearest neighbor as their embedding characteristics; otherwise, their embedding was set to a zero vector as their embedding characteristic.

### 2.5. Biological Function Information

The biological function of drugs is another important feature that can be represented by drug-related receptors. All of the biological function information can be fused by the multiple calculating similarity matrixes. In this work, *M* types of receptor bound to drugs are used to establish the adjacency matrixes with different dimension spaces. In each matrix, the values 1 or 0 indicate the existence or the inexistence of the drug-receptor relation, respectively, and the dimension is determined by the number of receptors. For example, there are 1660 receptors of the target type, so a drug can be embedded as a 1660-dimensional initial vector. Given a drug $D_{1}$ and a drug $D_{2}$, their initial embedding vectors are $d_{1}$ and $d_{2}$, and the similarity between them is based on the Euclidean distance and satisfies the formula as follows:(7)$$S(D_{1},D_{2})=\sqrt{\sum_{i=1}^{n}{\left(d_{1i}-d_{2i}\right)}^{2}}$$
where $d_{1i}$ and $d_{2i}$ are, respectively, the element representation of the vectors $d_{1}$ and $d_{2}$, and n means the number of dimensions. Therefore, the weak similarity matrices among these initial vectors are calculated in each function space type. Then, it is particularly important to integrate *M* weak similarity matrices into a strong similarity matrix. Similarity Network Fusion (SNF), proposed by Wang et al. [41], is a competent approach to fusing massive biological context similarities [18]. SNF is a non-linear similarity fusion method, combining multi-similarity matrices into a single integrated similarity matrix, which carries all appropriate representation information. Eventually, SNF is applied to fuse *M* similarity *N × N* matrix into one *N × N* matrix, where *N* is the number of drugs. Figure 4 reveals the details of this process.

### 2.6. Attentional Multi-Feature Integration and Prediction

To integrate the chemical sequence feature, the network structural distribution feature and the biological function feature for interaction prediction, the attentional multi-feature integration module is proposed. The attention [42] module can fuse multi-level features by using attention scores to reflect adaptive weights, namely, fusing each feature according to a score of feature significance level. Given the initial representation vectors θ and the attention enhance vectors $\tilde{\theta }$ can be obtained as the following function:(8)$$\tilde{\theta }=Attention(Q,K,V)=softmax(\frac{QK^{⊤}}{\sqrt{d_{k}}})V$$
where Q, K and V stand on behalf of the product of input vector and three matrices. The $d_{k}$ represents a normalization coefficient. Then, after calculating by a hidden layer, the classification result can be captured.

For the further screening of the efficient characteristic factors, before the three features were input into the attention module, a series of dimension-reducing processes were carried out. Firstly, we used CNN to reduce the feature matrix dimension of the chemical sequence, which contains 64 kernels. Inspired by NIN [43], we used a two-dimensional global average pooling, instead of the flattening operation. Then, the network structural feature vector was put into a fully connected layer containing 64 neurons, and the biological function features passed through a hidden layer of 300 neurons and a hidden layer with 64 neurons. Ultimately, after the elements’ weight of feature vectors were redistributed over attention, all of the processed features were put into the attention module. During the end-to-end training process, the Adam optimizer was used to adjust the learning rate automatically from 1 × 10^−3^. To avoid overfitting, we adopted the dropout rate of 0.3 on the hidden layers.

## 3. Experimental Results and Discussion

### 3.1. Evaluation Criteria

In this paper, classifying each pair of drugs as interaction or non-interaction, we utilized metrics that are frequently used in classification to assess the effectiveness and robustness of our model through distinct perspectives, including five parameters: Accuracy (*Acc*), Sensitivity (*Sen*), Precision (*Prec*), F-measure (*F*1) and Matthews’s Correlation Coefficient (*MCC*). These evaluation parameters are represented as follows:(9)$$Acc=\frac{TP+TN}{TN+TP+FN+FP}\;$$
(10)$$Prec=\frac{TP}{TP+FP}\;$$
(11)$$Sen=\frac{TP}{TP+FN}\;$$
(12)$$F1=\frac{2\times Prec\times Sen}{Prec+Sen}\;$$
(13)$$MCC=\frac{TP\times TN-FP\times FN}{\sqrt{\left(TP+FP)\times (TN+FN)\times (TN+FP)\times (TP+FN\right)}}\;$$
where true positive, false negative, true negative and false positive are, respectively, are represented by *TP*, *FN*, *TN* and *FP*.

Meanwhile, according to these parameters, we evaluated the performance of the proposed model by constructing an AUC, the area under the receiver operating characteristic (ROC) curves, and AUPR, which is the area under the precision-recall (PR) curve. AUPR represents an appropriate criterion if the count of negative and positive samples is not the same, just as suggested by [44]. A larger AUC shows a better predictor. We also exploited a five-fold cross-validation and the mean value to assess the performance of methods.

### 3.2. Assessment of Prediction Ability

In this experiment, to demonstrate the superior performance and robustness of BioChemDDI, we employed it on the four datasets. The jack-knife and q-fold cross-validation (CV) tests [45] are often used to examine whether the predictor is effective or not. We adopted the five-fold cross-validation to enhance the persuasion of the results, which are displayed in Table 2. The average values of AUC and AUPR can reach 0.9711 and 0.9701 or more in all four datasets. Specifically, on DS1, reaching the highest score of 0.9927 and 0.9921. In addition, the average scores of other evaluation criteria are also high; thus, the MCC of the four datasets gained the scores of 92.32, 81.48, 91.08 and 92.05. It can be concluded that the data scale produces an impact on model performance and results, but the worse performance, among the four datasets, also reached 90.72%, 89.54%, 92.21%, 90.86% and 81.48% on the evaluation criteria.

It is obvious that the proposed model has an excellent capability to identify positive and negative samples and predict novel DDIs. Furthermore, the higher AUC and AUPR show that our framework has a superior predictive performance, and the stability and the robustness of the proposed framework can be indicated by the lower variance of the results. As all of the evaluation results show, the performance can improve as the number of data increase. Four ROC curves and four PR curves were drawn as shown in Figure 5 and Figure 6. Finally, it can be concluded that the framework is effective and robust in drug–drug interaction prediction.

### 3.3. Ablation Experiments

A set of ablation experiments were conducted to validate the contribution levels of the biochemical and network structural information on DS1. The strategy of the training and the evaluation was similar to the procession of model training in Section 3.1. For convenience, we set the experimental number for each ablation experiment and Table 3 shows the results. For a fair comparison of each contribution level of feature, the attention machine should not be used, that is, comparing between (a), (b), (c) and (d). Without chemical sequence level, the values of AUC and ACPR declined by 0.0028 and 0.0031, respectively. Without network structure information, the performance decreased by 0.0135 and 0.0157 in terms of AUC and AUPR, respectively. Without biological function information, the AUC and AUPR were 0.0015 and 0.0019 lower than BioChemDDI, respectively. The experimental results show that the network structural information performs the most significant contribution. One of the possible reasons for this is that the representation pattern between known correlations can be viewed as direct features. Additionally, on the system constituted by abundant direct features, these features are more important than the other correlation features. Due to the important role of the graph structural feature in the proposed model, we discuss the related impact factors in Section 3.4 and Section 3.5.

Apart from this, we performed ablation studies to investigate the contributions of attentional integration at different features. As Table 3 illustrates, respectively comparing (a) with (A), (b) with (B), (c) with (C) and (d) with (D), the performance of the prediction dropped by 0.0073 and 0.0087, 0.0077 and 0.0095, 0.0024 and 0.0031, 0.0057 and 0.0066, respectively, in terms of AUC and ACPR, without attention redistribution. The ablation experiments show that the model can reach optimum performance only when all the information and attentional integration are used.

### 3.4. Influence of Graph Embedding Methods

The graph embedding method of HARP-LINE based on graph collapse was used in our experiment to extract information on the network structure. In order to test the impact of the graph embedding method on the performance of the model, we tested four embedding methods containing LINE [40], Deepwalk [46], HARP-Deepwalk and Laplace Eigenmaps [47] on the four datasets. All the parameters of each embedding method were default, and the embedding dimension was set to 64, the same as in the case of HARP-LINE. To reduce the influence of biochemical features on the comparison experiment, the attention module is not available. From Figure 7, we can see that, although there are a few fluctuations in the evaluation criteria, from the overall trend, HARP-LINE obtained a better performance. That is attributed to HARP-LINE being able to obtain deep network structural information by focusing on long-distance embedding configurations.

### 3.5. Influence of the Embedding Dimension of the Graph Structural Feature

Embedding dimension has a crucial influence on the learning ability of graph structural information, in which a large value could cause inefficiency for training, while a small value could limit the learning ability. When generating the feature vectors of the network structure, we set the embedding dimension to 64 to obtain the rich topology information. To test the influence of the graph embedding dimension on the model performance, a comparative experiment was implemented. In this section, we analyzed the performance in different embedding dimensions containing 16, 32, 64 and 128. As Figure 8 shows, when the dimension is set to 64, there is a significant promotion of all the evaluation indicators on DS1 and D-DS3. Additionally, the competitive results were also obtained on the other datasets. Thus, we set the dimension of graph embedding as 64.

### 3.6. Comparison with Various Classifier Models

In the BioChemDDI model, we identified the potential DDIs through the feedforward neural network with all feature information. To verify the influence of the attentional neural network on the model performance, we compared the effects of various classifiers. Specifically, we kept the feature descriptors unchanged andused Gaussian NB (GNB), Decision Tree (DT) and Logistic Regression (LR) to implement the prediction task. For intuitive comparison, we displayed the experimental results in the form of a histogram in Figure 9. It can be easily seen that BioChemDDI shows a clear predominance on all evaluation criteria. These results demonstrate that the performance of our method can be enhanced with an attentional neural network.

### 3.7. Comparison with Other Methods

In order to comprehensively evaluate the performance of our proposed model, seven computational methods were chosen as baseline to make the comparison. As mentioned above, we utilized DS2, usually used to test model performance, as the comparison dataset. We compared models containing NDD [18], ISCMF [48], DPDDI [37], GCN-BMP [49], AttentionDDI [50], BioDKG-DDI [31] and an ensemble model [33]. These methods use the dataset of DS2 and five-fold cross-validation. We selected all of the reported evaluation criteria of each comparison method in this paper. The comparison results of the evaluation measure are shown in Table 4, which shows that BioChemDDI reaches the best performance with the outstanding improvement of 0.0043~0.2521, 0.0083~0.1856, 0.0174, 0.0043~0.0721, 0.0299~0.1631 regarding *Sen*, *F*1, *MCC*, AUC and AUPR. The possible reason is that the BioChemDDI model can fully learn the network structural information and extract more effective information through the attention mechanism integrating biochemical information. Nevertheless, in terms of *Acc* and *Prec*, our model was lower than the Ensemble Model and ISCMF. On the one hand, the Ensemble Model has an ensemble classifier, which obtains accurate classification results by classifier voting. On the other hand, ISCMF obtained a higher value of *Prec*, but a lower value of F1, because it only ensures the ability of predicting positive samples. On the whole, although BioChemDDI is a powerful method, there is still room for further improvement.

### 3.8. Case Studies: Cyclophosphamide, Regorafenib and Allopurinol

To further estimate the ability of BioChemDDI to predict novel DDIs, we conducted case studies on the drugs cyclophosphamide, regorafenib and allopurinol, which are associated with three cancers: breast cancer, hepatocellular carcinoma and malignancies. In the experiment, all 219,247 known drug–drug pairs, among the 1940 drugs, were used to train the prediction model, and all the remaining pairs related to these three test drugs were used as the test set to predict the existence of DDIs; in other words, the test pairs were not included in the train set. The positive samples in the test set were unvisualizable to our model and we predicted the unvisualizable DDIs, which can be seen as unknown DDI, through learning visualizable samples. Finally, the prediction of DDIs can be validated by these positive samples not known beforehand. In particular, we used BioChemDDI to identify all unknown interactions among the three drugs and ranked them according to the prediction score and then searched the DrugBank to verify our results. The top 30 ranked prediction results for each pair of cancer-related drugs are shown in Table 5, Table 6 and Table 7.

The results of the experiment show that 24, 18 and 20 interactions were confirmed by DrugBank with the top five predictions confirmed, except *cyclophosphamide-amoxicillin*. Unconfirmed DDIs should be further verified through the wet lab. Especially, the drug *amoxicillin* may potentially interact with *Cyclophosphamide* with high confidence. To further illustrate the potential interaction mechanism between cyclophosphamide and amoxicillin, we consulted their relevant pharmacokinetic information in DrugBank. We found that cyclophosphamide is the substrate and the inducer of the Cytochrome P450 2C8 enzyme, and meanwhile, amoxicillin is the inhibitor of the Cytochrome P450 2C8 enzyme, which means amoxicillin may impact the metabolism process of cyclophosphamide. To clearly observe the results of the identification of potential DDIs for these three cancer-related drugs, we visualized the newly discovered and known interactions of the top 30 in Figure 10. In addition, the full prediction scores in each case study are reported in Appendix A.

## 4. Conclusions

During the development of drugs and their clinical application, combining drugs to treat complex diseases may induce adverse reactions, which makes the identification of potential DDIs before clinical medications are administered more and more urgent. In this paper, we studied how to exploit network topology structure and biochemical information to predict potential DDIs. Additionally, we proposed a novel model, named BioChemDDI, to adaptively learn drug features from the chemical sequence information, network structural information and biological function information, and fused them to enhance features through the attention module. After sufficient supervised learning, the model accurately predicted potential DDIs. On the one hand, the case studies of three cancer-related drugs indicate the good prediction ability of our model. On the other hand, our model could be seen as a pre-screening tool for potential DDIs. In this way, the workload of exploring the unknown complex interactions of drugs can be reduced. In the future, to improve our framework, choosing negative samples with a more reasonable way to reduce the noise brought by unbalancing the original dataset and transferring our framework to predict interactions between unknown drugs will be considered.

## 5. Patents

The computational platform web server was built and is accessible at: http://120.77.11.78/BioChemDDI/ (accessed on 11 April 2022).

## Figures and Tables

**Figure 1 biology-11-00758-f001:**
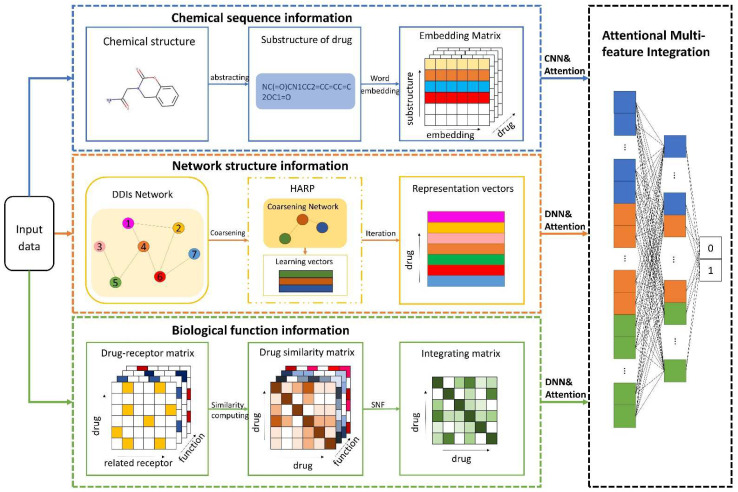
The pipeline of BioChemDDI. After inputting data, the sub-model of chemical sequence information obtains an embedding matrix of each drug that can be reduced as a 1-dimension vector by the CNN. The network structural information is learned by the method of HARP, and the feature vector can be further screened for the characteristic factors through the hidden layers. Multiple biological function matrices are used to generate multiple drug similarity matrices, which can be fused by SNF and further screened for characteristic factors by the hidden layers. The module of feature integration with attention contains three hidden layers that finish the final DDI prediction.

**Figure 2 biology-11-00758-f002:**
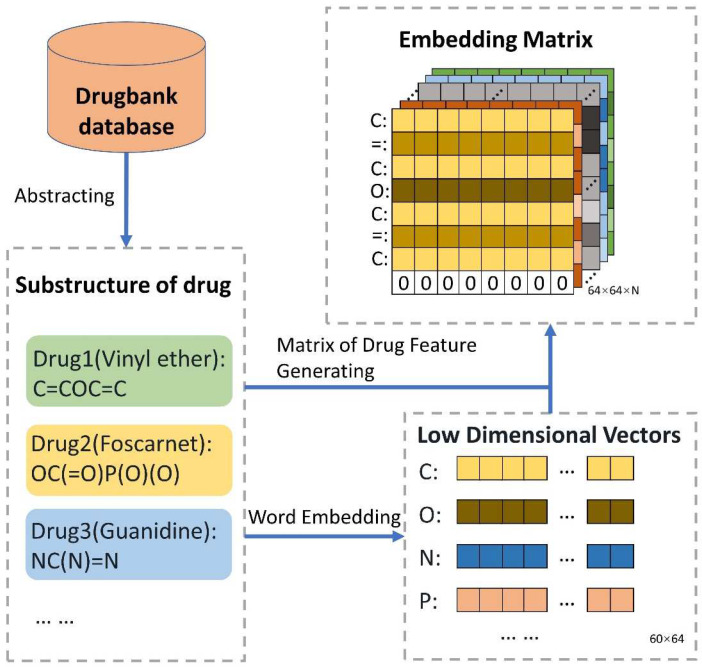
The guideline for extracting features of chemical sequence. The SMILES of all drugs were obtained from DrugBank. Additionally, vector representation, the matrix of 60 × 64, of all 60 biological symbols is learned by word2vec. Then, the SMILES of each drug corresponds to its own symbolic vector representation. Finally, N drugs can be indicated as N matrixes. To ensure the identical scale of the drug matrix, the strategies were adopted as follows: a. If the sequence length of the drug is shorter than 64, the vector of zero is attached to the drug matrix; b. If the sequence length of the drug is longer than 64, it is reduced to 64.

**Figure 3 biology-11-00758-f003:**
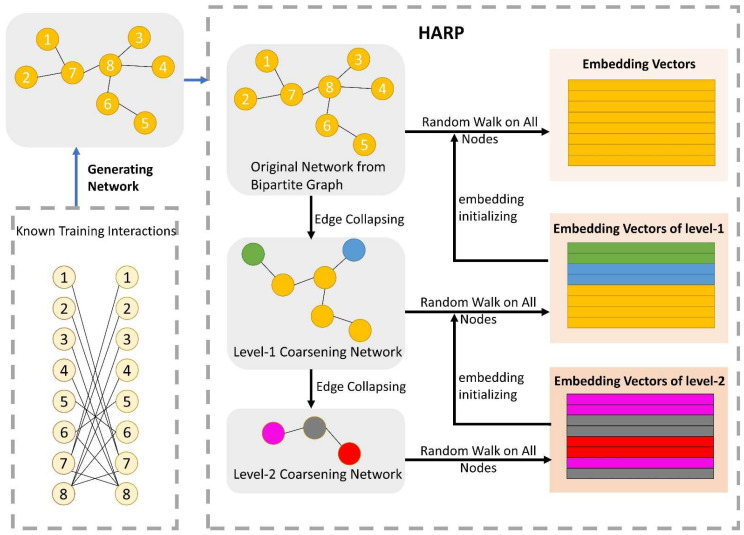
The process of the network representation learning module. There are three parts to HARP: graph coarsening, graph embedding and representation refining. After generating the drug network from the training of the positive samples, the original network is folded continuously to gain many coarse subnetworks. For example, in the level-1 coarsening network, according to the folded rule, node 1 and node 2 are seen as a whole sharing the same embedding vector, when extracting embedding vectors of nodes in the level-1 coarsening network. The final embedding vectors of the lower-level subnetwork are exploited as the initial vector of its superior level subnetwork. Iterating upward constantly, ending at the original graph, and finally obtaining all of the node embedding vectors, which have rich structural information. The LINE method is utilized to obtain embedding vectors for the nodes of each subnetwork.

**Figure 4 biology-11-00758-f004:**
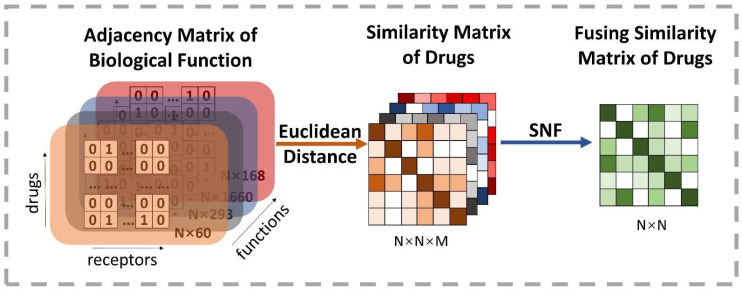
Illustration of the extracting process of the fusion drug similarity matrix. At first, *M* types of biological function information are used to construct adjacency matrixes, whose row indicates the drug’s initial vector and the number of receptors decides the dimension. Thus, each row is regarded as a high-dimensional space representation of drugs. Then, the drug similarity matrix based on different kinds of space can be obtained by calculating the Euclidean distance on each pairwise embedding vector of drugs. Finally, it exploits the SNF method to produce a comprehensive representation of the biological function information by fusing similarity matrixes of all types of space.

**Figure 5 biology-11-00758-f005:**
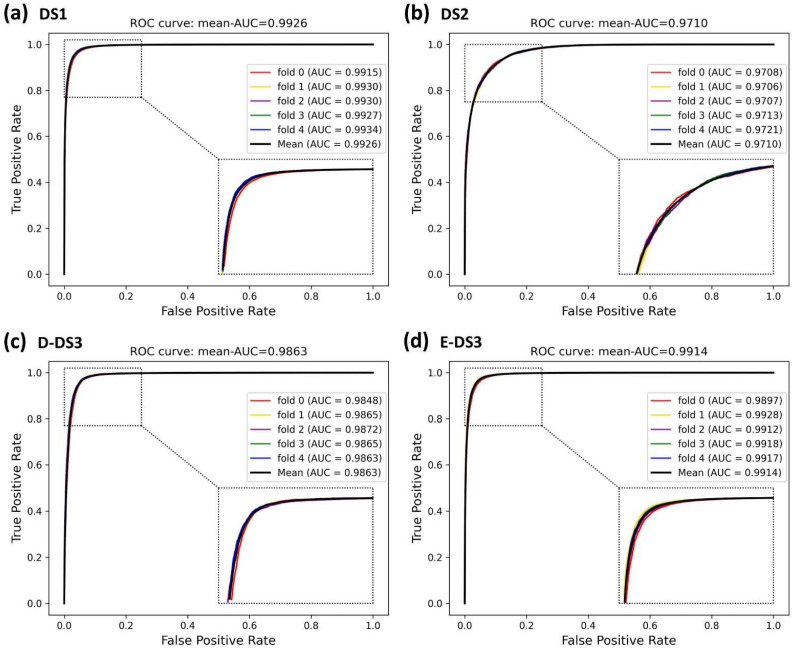
ROC curves of the five-fold cross-validation on the four datasets by the proposed method, BioChemDDI.

**Figure 6 biology-11-00758-f006:**
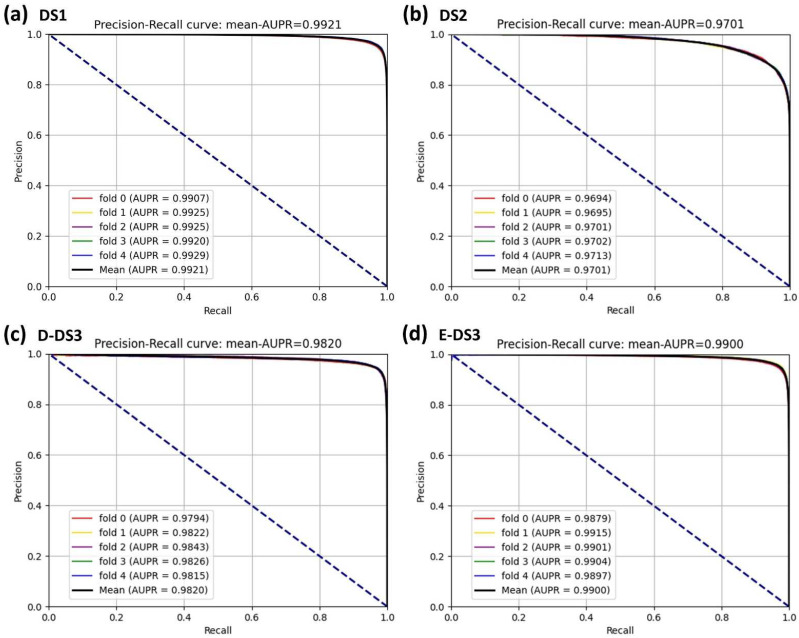
PR curves of the five-fold cross-validation on the four datasets by the proposed method, BioChemDDI.

**Figure 7 biology-11-00758-f007:**
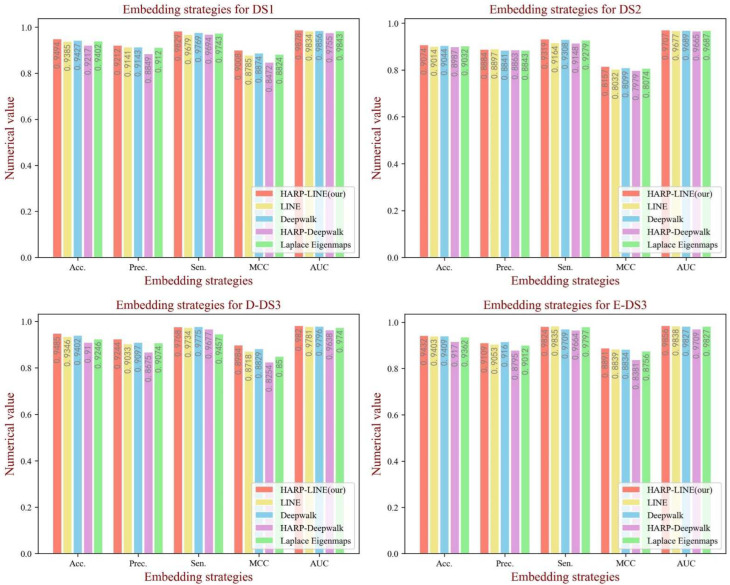
The performance of different graph embedding methods on each dataset.

**Figure 8 biology-11-00758-f008:**
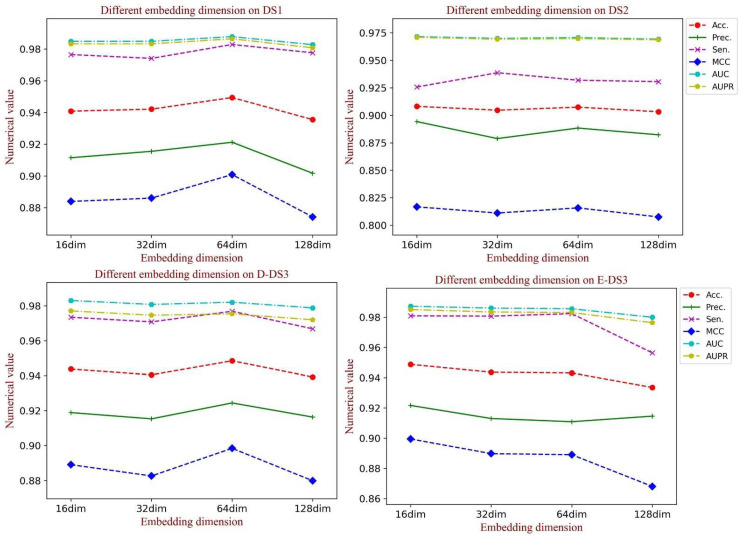
The performance of different dimensions of the graph embedding vector on each dataset.

**Figure 9 biology-11-00758-f009:**
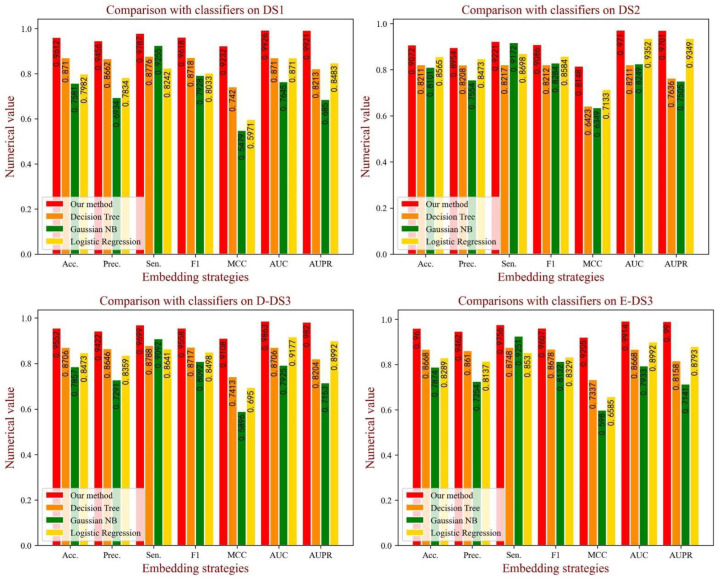
The performance of different classifiers on each dataset.

**Figure 10 biology-11-00758-f010:**
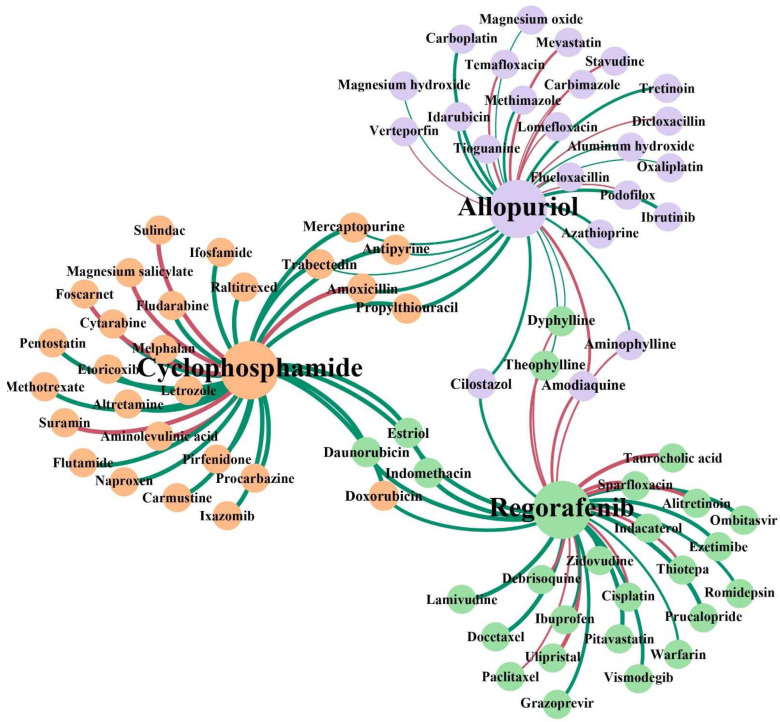
Network visualization of the newly discovered and known DDIs of the top 30 in each case study. The three colors of the node represent the three case studies. The red lines represent the interactions with no support and the green lines represent confirmed interactions. The thickness of the line between any two nodes represents the prediction score of the interaction.

**Table 1 biology-11-00758-t001:** The feature description and interaction details of the four datasets.

Datasets Name	Number of Drugs	Number of Pairs	Number of Interactions	Biological Function Information
DS1 [31,32]	1940	3,763,600	219,247	Carrier, Target, Enzyme, Transporter
DS2 [33]	548	300,304	97,168	Pathway, Target, Enzyme, Indication, Offside, Side effect, Transporter
D-DS3 [27]	1562	2,439,844	55,278	Target, Carrier, Enzyme, Transporter, Molecular fingerprint
E-DS3 [27]	1562	2,439,844	125,298	Carrier, Target, Enzyme, Transporter, Molecular fingerprint.

**Table 2 biology-11-00758-t002:** Five-fold cross-validation results through BioChemDDI.

Datasets	Fold	*Acc*	*Prec*	*Sen*	*F*1	*MCC*	AUC	AUPR
DS1	0	0.9583	0.9436	0.9748	0.9589	0.9170	0.9915	0.9907
	1	0.9619	0.9469	0.9787	0.9626	0.9244	0.9930	0.9925
	2	0.9616	0.9458	0.9793	0.9622	0.9237	0.9930	0.9925
	3	0.9610	0.9449	0.9792	0.9617	0.9227	0.9927	0.9920
	4	0.9634	0.9469	0.9817	0.9640	0.9273	0.9934	0.9929
	**Average**	**0.9612 ± 0.0019**	**0.9456 ± 0.0014**	**0.9787 ± 0.0025**	**0.9618 ± 0.0019**	**0.9232 ± 0.0038**	**0.9926 ± 0.0007**	**0.9921 ± 0.0009**
DS2	0	0.9101	0.8991	0.9237	0.9113	0.8204	0.9708	0.9694
	1	0.9063	0.8964	0.919	0.9075	0.8130	0.9706	0.9695
	2	0.9051	0.8913	0.9227	0.9067	0.8106	0.9707	0.9701
	3	0.9073	0.8931	0.9253	0.9089	0.8151	0.9713	0.9702
	4	0.9073	0.8973	0.9198	0.9084	0.8148	0.9721	0.9713
	**Average**	**0.9072 ± 0.0018**	**0.8954 ± 0.0032**	**0.9221 ± 0.0027**	**0.9086 ± 0.0017**	**0.8148 ± 0.0036**	**0.9711 ± 0.0006**	**0.9701 ± 0.0008**
D-DS3	0	0.9545	0.9370	0.9745	0.9554	0.9097	0.9848	0.9794
	1	0.9549	0.9385	0.976	0.9557	0.9105	0.9865	0.9822
	2	0.9539	0.9515	0.9565	0.9540	0.9078	0.9872	0.9843
	3	0.9559	0.9408	0.9730	0.9566	0.9123	0.9865	0.9826
	4	0.9567	0.9432	0.9719	0.9573	0.9138	0.9863	0.9815
	**Average**	**0.9552 ± 0.0011**	**0.9422 ± 0.0057**	**0.9699 ± 0.0076**	**0.9558 ± 0.0013**	**0.9108 ± 0.0023**	**0.9863 ± 0.0009**	**0.9820 ± 0.0018**
E-DS3	0	0.9551	0.9360	0.9771	0.9561	0.9111	0.9897	0.9879
	1	0.966	0.9511	0.9773	0.9641	0.9275	0.9928	0.9915
	2	0.9586	0.9469	0.9718	0.9592	0.9176	0.9912	0.9901
	3	0.9615	0.9475	0.9771	0.9621	0.9235	0.9918	0.9904
	4	0.9613	0.9493	0.9745	0.9618	0.9228	0.9917	0.9897
	**Average**	**0.9600 ± 0.0033**	**0.9462 ± 0.0059**	**0.9756 ± 0.0024**	**0.9607 ± 0.0031**	**0.9205 ± 0.0063**	**0.9914 ± 0.0011**	**0.9900 ± 0.0013**

**Table 3 biology-11-00758-t003:** Results of the ablation on DS1.

	Experimental Number	Chemical Sequence	Network Structure	Biological Function	AUC	AUPR
Without the attention mechanism	(a)	F	T	T	0.9842	0.9824
	(b)	T	F	T	0.9735	0.9698
	(c)	T	T	F	0.9855	0.9836
	(d)	T	T	T	0.9870	0.9855
With the attention mechanism	(A)	F	T	T	0.9915	0.9911
	(B)	T	F	T	0.9812	0.9793
	(C)	T	T	F	0.9879	0.9867
	(D)	T	T	T	**0.9927**	**0.9921**

F means False; T means True and with attention. The bold values indicate the highest values of AUC and AUPR.

**Table 4 biology-11-00758-t004:** Comparison of our proposed method with seven computational methods.

Method	*Acc*	*Prec*	*Sen*	*F*1	*MCC*	AUC	AUPR
NDD [18]	(-)	0.8330	0.8360	0.8350	(-)	0.9540	0.9220
ISCMF [48]	0.8510	**0.9880**	0.8510	0.8850	(-)	0.8990	0.8640
DPDDI [37]	0.9400	0.7540	0.8100	0.8400	(-)	0.9560	0.9070
GCN-BMP [49]	(-)	(-)	(-)	0.8500	(-)	0.9666	0.9402
AttentionDDI [50]	(-)	(-)	(-)	(-)	(-)	0.9540	0.9240
BioDKG-DDI [31]	0.8984	0.8835	0.9178	0.9003	0.7974	0.9668	(-)
Ensemble Model [33]	**0.9550**	0.7850	0.6700	0.7230	(-)	0.9570	0.8070
Proposed Method	0.9072	0.8954	**0.9221**	**0.9086**	**0.8148**	**0.9711**	**0.9701**

The values in bold indicate the highest values in each column. The symbol (-) indicates the evaluation criteria are not reported in the original articles.

**Table 5 biology-11-00758-t005:** The top 30 interactions of *Cyclophosphamide*.

Rank	Drug Name	DrugBank ID	Evidence	Rank	Drug Name	DrugBank ID	Evidence
1	*Altretamine*	DB00488	Confirmed	16	*Fludarabine*	DB01073	Confirmed
2	*Amoxicillin*	DB01060	N.A.	17	*Naproxen*	DB00788	Confirmed
3	*Ifosfamide*	DB01181	Confirmed	18	*Mercaptopurine*	DB01033	Confirmed
4	*Ixazomib*	DB09570	Confirmed	19	*Propylthiouracil*	DB00550	Confirmed
5	*Indomethacin*	DB00328	Confirmed	20	*Sulindac*	DB00605	N.A.
6	*Aminolevulinic acid*	DB00855	N.A.	21	*Doxorubicin*	DB00997	Confirmed
7	*Magnesium salicylate*	DB01397	N.A.	22	*Foscarnet*	DB00529	N.A.
8	*Cytarabine*	DB00987	Confirmed	23	*Suramin*	DB04786	N.A.
9	*Pirfenidone*	DB04951	Confirmed	24	*Antipyrine*	DB01435	Confirmed
10	*Melphalan*	DB01042	Confirmed	25	*Methotrexate*	DB00563	Confirmed
11	*Estriol*	DB04573	Confirmed	26	*Etoricoxib*	DB01628	Confirmed
12	*Pentostatin*	DB00552	Confirmed	27	*Letrozole*	DB01006	Confirmed
13	*Daunorubicin*	DB00694	Confirmed	28	*Flutamide*	DB00499	Confirmed
14	*Raltitrexed*	DB00293	Confirmed	29	*Procarbazine*	DB01168	Confirmed
15	*Carmustine*	DB00262	Confirmed	30	*Trabectedin*	DB05109	Confirmed

**Table 6 biology-11-00758-t006:** The top 30 interactions of *Regorafenib*.

Rank	Drug Name	DrugBank ID	Evidence	Rank	Drug Name	DrugBank ID	Evidence
1	*Indomethacin*	DB00328	Confirmed	16	*Doxorubicin*	DB00997	Confirmed
2	*Pitavastatin*	DB08860	Confirmed	17	*Grazoprevir*	DB11575	Confirmed
3	*Lamivudine*	DB00709	Confirmed	18	*Romidepsin*	DB06176	Confirmed
4	*Daunorubicin*	DB00694	Confirmed	19	*Amodiaquine*	DB00613	N.A.
5	*Prucalopride*	DB06480	Confirmed	20	*Cilostazol*	DB01166	Confirmed
6	*Docetaxel*	DB01248	Confirmed	21	*Debrisoquine*	DB04840	N.A.
7	*Estriol*	DB04573	Confirmed	22	*Thiotepa*	DB04572	N.A.
8	*Taurocholic acid*	DB04348	N.A.	23	*Cisplatin*	DB00515	N.A.
9	*Ulipristal*	DB08867	N.A.	24	*Ibuprofen*	DB01050	Confirmed
10	*Ombitasvir*	DB09296	Confirmed	25	*Theophylline*	DB00277	N.A.
11	*Ezetimibe*	DB00973	Confirmed	26	*Warfarin*	DB00682	Confirmed
12	*Zidovudine*	DB00495	Confirmed	27	*Indacaterol*	DB05039	Confirmed
13	*Sparfloxacin*	DB01208	N.A.	28	*Paclitaxel*	DB01229	N.A.
14	*Alitretinoin*	DB00523	N.A.	29	*Aminophylline*	DB01223	N.A.
15	*Vismodegib*	DB08828	Confirmed	30	*Dyphylline*	DB00651	N.A.

**Table 7 biology-11-00758-t007:** The top 30 interactions of *Allopurinol*.

Rank	Drug Name	DrugBank ID	Evidence	Rank	Drug Name	DrugBank ID	Evidence
1	*Propylthiouracil*	DB00550	Confirmed	16	*Flucloxacillin*	DB00301	N.A.
2	*Amoxicillin*	DB01060	Confirmed	17	*Antipyrine*	DB01435	Confirmed
3	*Ibrutinib*	DB09053	Confirmed	18	*Dicloxacillin*	DB00485	N.A.
4	*Carboplatin*	DB00958	Confirmed	19	*Trabectedin*	DB05109	Confirmed
5	*Tretinoin*	DB00755	Confirmed	20	*Podofilox*	DB01179	N.A.
6	*Amodiaquine*	DB00613	N.A.	21	*Tioguanine*	DB00352	Confirmed
7	*Cilostazol*	DB01166	Confirmed	22	*Lomefloxacin*	DB00978	N.A.
8	*Azathioprine*	DB00993	Confirmed	23	*Aluminum hydroxide*	DB06723	Confirmed
9	*Mevastatin*	DB06693	N.A.	24	*Theophylline*	DB00277	Confirmed
10	*Methimazole*	DB00763	Confirmed	25	*Magnesium hydroxide*	DB09104	Confirmed
11	*Idarubicin*	DB01177	Confirmed	26	*Oxaliplatin*	DB00526	Confirmed
12	*Stavudine*	DB00649	N.A.	27	*Dyphylline*	DB00651	Confirmed
13	*Mercaptopurine*	DB01033	Confirmed	28	*Verteporfin*	DB00460	N.A.
14	*Aminophylline*	DB01223	Confirmed	29	*Magnesium oxide*	DB01377	Confirmed
15	*Temafloxacin*	DB01405	N.A.	30	*Carbimazole*	DB00389	N.A.

## Data Availability

BioChemDDI is publicly available as an online predictor at: http://120.77.11.78/BioChemDDI/ (accessed on 11 April 2022). The original data of the drugs are available at: https://go.drugbank.com/ (accessed on 3 June 2021).

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
