# Peer review of "BioChemDDI: Predicting Drug–Drug Interactions by Fusing Biochemical and Structural Information through a Self-Attention Mechanism"

_biology, 2022, doi:10.3390/biology11050758_

Round 1

Reviewer 1 Report

Ren et al. are presenting their application of predicting drug-drug interactions using machine learning. The topic of DDI is indeed an essential area of research that deserves a good amount of attention. However, there are several major concerns may impede the acceptance of this manuscript.

First of all, the English language in this paper gives the reviewer a hard time to perceive authors' intended meaning. It can be a fair request that authors should carefully revisit their writing before making the next submission.

Another concern is that experimental DDI validation should not be downplayed in the section of Introduction. No matter how the prediction is performed, the experimental validation is the ultimate way to assess whether there is DDI or not. The computational prediction can help provide a direction or prioritize the experimental tests, but not independently function as an indicator for DDI.

The third point is regarding the so-called "case study". The case study is merely the pure prediction without any experimental validations at all. If a DDI is already reported and can be confirmed on DrugBank, how could authors claim that DDI is "unknown"? BioChemDDI has a vital feature that it incorporates drug-receptor/biological functional information into the training process. Then the reviewer would expect that in the case study, not only DDI drug pairs but also potential mechanism for that DDI should be revealed. It will be significant to help jump over the mud season in Vermont. 

This manuscript is slightly below the border line.

Author Response

Reviewer: 1

Comments and Suggestions for Authors:

Ren et al. are presenting their application of predicting drug-drug interactions using machine learning. The topic of DDI is indeed an essential area of research that deserves a good amount of attention. However, there are several major concerns may impede the acceptance of this manuscript.

Response:

The authors greatly appreciate your time and efforts. Thank you for your recognition of our work and valuable comments. There is no doubt that this will be of great help to us to improve our work. We seriously consider your valuable questions and make revisions based on your revision opinions point by point. We sincerely hope that our work can be published in the future after careful revisions. Individual response to each question is answered below.

Comment 1:

First of all, the English language in this paper gives the reviewer a hard time to perceive authors' intended meaning. It can be a fair request that authors should carefully revisit their writing before making the next submission.

Response 1:

We are grateful for the suggestion. We have revised the whole manuscript carefully and tried to avoid any grammar or syntax errors. In addition, we have asked several colleagues who are skilled authors of English language papers to check the English. We believe that the language is now acceptable for the review process. Moreover, for understanding our work easily, we added some explanatory sentences to Methods section to improve the readability of our manuscript.

Comment 2:

Another concern is that experimental DDI validation should not be downplayed in the section of Introduction. No matter how the prediction is performed, the experimental validation is the ultimate way to assess whether there is DDI or not. The computational prediction can help provide a direction or prioritize the experimental tests, but not independently function as an indicator for DDI.

Response 2:

Thank you for your very helpful and constructive comments. Just as the reviewer said, the computational prediction can help provide a direction or prioritize the experimental tests, but not independently function as an indicator for DDI. However, with increasing knowledge of biological structures, computer-aided drug discovery tools are getting a lot of attention in the pharmaceutical industry and academia. In the revised manuscript, we fully introduced the process of the experimental DDI validation and several strategies for predicting potential DDIs. We emphasized the importance of experimental validation and described the shortcomings of the experiments based on pharmacokinetics. Meanwhile, the necessity of utilizing computational methods as a prescreening tool was also elaborated. Like the sentences as follows:

‘Alterations in drug pharmacokinetics and drug pharmacodynamics can be caused by DDIs [6]. Pharmacokinetic DDI crops up when the perpetrator drug disrupts the absorption, distribution, metabolism, and elimination (ADME) of the victim drug, and pharmacodynamic DDI crops up when the perpetrator drug interacts with the protein of the victim drug or other protein within the same signaling pathway [4]. To screen and analyze the unknown DDIs, biological techniques are mainly used, which are regarded as the ultimate way to judge and validate the DDIs, containing metabo-lism-based DDIs and transporter-based DDIs. Such as testing the drug whether is the inhibitor/inducer or substrate of CYP enzymes [7], and testing the drug whether is the inhibitor/inducer or substrate of P-gp transporter [3]. Then, based on the vitro parame-ters, the cumbersome dynamic model (e.g., PBPK) and expensive vivo experiment should be constructed and analyzed for the final validation.’

Comment 3:

The third point is regarding the so-called "case study". The case study is merely the pure prediction without any experimental validations at all. If a DDI is already reported and can be confirmed on DrugBank, how could authors claim that DDI is "unknown"? BioChemDDI has a vital feature that it incorporates drug-receptor/biological functional information into the training process. Then the reviewer would expect that in the case study, not only DDI drug pairs but also potential mechanism for that DDI should be revealed. It will be significant to help jump over the mud season in Vermont.

Response 3:

We totally understand the reviewer's concern and thank you for your rigorous consideration and your suggestion of revealing the potential mechanism for DDI. First, the experimental validation is meaningful work. The case study methodology serves to provide a framework for the evaluation and analysis of complex issues, which aims to illustrate the performance of our model.

The work mainly aims to propose a computational model for preliminary screening out high confidence DDIs on a large-scale level like the previous works [1-6]. But as soon as we get enough funding, we will further perform wet experiments. Additionally, we analyzed the potential mechanism of the predicted DDI to validate the results. Second, we revised some sentences which were described vaguely. Specifically, the positive samples in the test set are unvisualizable to our model and we predict the unvisualizable DDI, which can be seen as unknown DDI, through learning visualizable samples. Finally, the prediction of DDIs can be validated by these positive samples not known beforehand. Third, we analyzed the potential mechanism of the drug pairs which has high confidence, as follows:

‘To further illustrate the potential interaction mechanism between Cyclophosphamide and Amoxicillin, we consult the relevant pharmacokinetics knowledge of them from DrugBank. We find that Cyclophosphamide is the substrate and the inducer of the Cy-tochrome P450 2C8 enzyme, and meanwhile, Amoxicillin is the inhibitor of the Cyto-chrome P450 2C8 enzyme, which means Amoxicillin may impact the metabolism process of Cyclophosphamide.’

  1. Deepika, S.; Geetha, T. A meta-learning framework using representation learning to predict drug-drug interaction. Journal of biomedical informatics 2018, 84, 136-147.
  2. Shi, J.-Y.; Huang, H.; Li, J.-X.; Lei, P.; Zhang, Y.-N.; Dong, K.; Yiu, S.-M. TMFUF: a triple matrix factorization-based unified framework for predicting comprehensive drug-drug interactions of new drugs. BMC bioinformatics 2018, 19, 27-37.
  3. Zhang, W.; Jing, K.; Huang, F.; Chen, Y.; Li, B.; Li, J.; Gong, J. SFLLN: a sparse feature learning ensemble method with linear neighborhood regularization for predicting drug–drug interactions. Information Sciences 2019, 497, 189-201.
  4. Zhang, W.; Chen, Y.; Li, D.; Yue, X. Manifold regularized matrix factorization for drug-drug interaction prediction. Journal of biomedical informatics 2018, 88, 90-97.
  5. Yu, H.; Mao, K.-T.; Shi, J.-Y.; Huang, H.; Chen, Z.; Dong, K.; Yiu, S.-M. Predicting and understanding comprehensive drug-drug interactions via semi-nonnegative matrix factorization. BMC systems biology 2018, 12, 101-110.
  6. Shi, J.-Y.; Mao, K.-T.; Yu, H.; Yiu, S.-M. Detecting drug communities and predicting comprehensive drug–drug interactions via balance regularized semi-nonnegative matrix factorization. Journal of cheminformatics 2019, 11, 1-16.

Reviewer 2 Report

biology-1702085, BioChemDDI: Predicting Drug-Drug Interactions by Fusing Biochemical Information and Structural Information through Self-Attention Mechanism

The manuscript presents in my opinion a very solid research. The paper is well written and edited. I found no significant problems to mention. My only concern is the choice of the journal and its readers. The manuscript would fit better a journal like Molecules or other journals that are focused on in silico prediction studies. It is difficult for a biology oriented reader to understand most of the chemical and mathematical concepts presented here. My only advice is that the authors try to detail and make more accessible the information for a public that is not familiar with this field. The authors could highlight the practical use of the application and focus more on its use.

Author Response

Reviewer: 2

Comments and Suggestions for Authors:

biology-1702085, BioChemDDI: Predicting Drug-Drug Interactions by Fusing Biochemical Information and Structural Information through Self-Attention Mechanism

The manuscript presents in my opinion a very solid research. The paper is well written and edited. I found no significant problems to mention. My only concern is the choice of the journal and its readers. The manuscript would fit better a journal like Molecules or other journals that are focused on in silico prediction studies. It is difficult for a biology oriented reader to understand most of the chemical and mathematical concepts presented here. My only advice is that the authors try to detail and make more accessible the information for a public that is not familiar with this field. The authors could highlight the practical use of the application and focus more on its use.

Response:

The authors greatly appreciate the time and efforts of the reviewer. Thank you for your helpful comments and for giving us such a high evaluation. In our manuscript, our major work is to predict drug-drug interactions (DDIs), which includes not only biological work, but also computer-based research. Identification of DDIs is helpful for promoting related biological research and our work can prescreen candidate DDIs for further vitro-vivo experiments. Then, for understanding our work easily, we fully introduce the process of the traditional experimental DDI validation and several strategies for predicting potential DDIs in the revised manuscript, as follows:

‘Drugs play a crucial role in curing diseases and enhancing the quality of life [1]. During drug development, drug-drug interactions (DDIs) is a critical consideration and the drug targeting the selected protein should be bioavailable (e.g., favorable absorption and metabolism) [2]. However, potential DDIs may lead to a strong rise or drop in plasma concentration of drug or metabolite, and even generate toxic compounds [3]. From a clinical perspective, a combination of drugs is used for the treatment of complex diseases, but unexpected DDIs may induce adverse reactions, which can give rise to drug withdrawal even to the death of the patient [4,5]. Thus, early identification of potential DDIs is very critical for drug development and medical safety.

       Alterations in drug pharmacokinetics and drug pharmacodynamics can be caused by DDIs [6]. Pharmacokinetic DDI crops up when the perpetrator drug disrupts the absorption, distribution, metabolism, and elimination (ADME) of the victim drug, and pharmacodynamic DDI crops up when the perpetrator drug interacts with the protein of the victim drug or other protein within the same signaling pathway [4]. To screen and analyze the unknown DDIs, biological techniques are mainly used, which are regarded as the ultimate way to judge and validate the DDIs, containing metabolism-based DDIs and transporter-based DDIs. Such as testing the drug whether is the inhibitor/inducer or substrate of CYP enzymes [7], and testing the drug whether is the inhibitor/inducer or substrate of P-gp transporter [3]. Then, based on the vitro parameters, the cumbersome dynamic model (e.g., PBPK) and expensive vivo experiment should be constructed and analyzed for the final validation.’

we rewrote some complex sentences that are difficult to understand to improve the readability of our manuscript. We highlighted the practical use of our model and added description information, which can make the readers easily understand our manuscript. We believe that the description is now more friendly for readers who are not familiar with this field. Additionally, We have revised the whole manuscript carefully and tried to avoid any grammar or syntax errors. We firmly believe that both biologists and computer scientists can easily understand our work.

Reviewer 3 Report

The paper, entitled "BioChemDDI: Predicting Drug-Drug Interactions by Fusing Biochemical Information and Structural Information through Self-Attention Mechanism", presents results demonstrate that BioChemDDI is a useful model to predict DDIs and can provide reliable candidates for biological experiments.The paper is well-organized and comprehensive. Although the inhibitor activity results are moderate, the used synthetic procedures are usefull for readers. The presented manuscript should be accepted after extensive editing of the language. This Manuscript is well written and needs to minor revision.

Comments:

  1. Every section of the manuscript must be written scientifically according to the published literature with appropriate references.
  2. The logical flow of this manuscript is not perfect. The authors have written several matters haphazardly. The work appears as groundwork. Spacing, punctuation marks, grammar, and spelling errors should be reviewed wholly.
  3. The flow of the introduction is not complete and unspecific. My recommendation is to construct the sentences more lucid and legible for more productive comprehension.
  4. The first paragraph of the introduction section contains no new information. Need to change.
  5. Author studied how to exploit network topology structure information and biochemical information to predict potential DDIs. On the one hand, case studies of three cancer-related drugs can indicate the good prediction ability of our model. On the other hand, our model could be seen as a pre-screening tool for potential DDIs. In this way, the work load of exploring unknown complex interactions of drugs can be reduced. In the future, to improve our framework, choosing negative samples with a more reasonable way to reduce the noise brought by unbalancing the original dataset and transferring our framework to predict interactions between unknown drugs would be considered. Is database screening possible for this model?

Author Response

Reviewer: 3

Comments and Suggestions for Authors:

The paper, entitled "BioChemDDI: Predicting Drug-Drug Interactions by Fusing Biochemical Information and Structural Information through Self-Attention Mechanism", presents results demonstrate that BioChemDDI is a useful model to predict DDIs and can provide reliable candidates for biological experiments.The paper is well-organized and comprehensive. Although the inhibitor activity results are moderate, the used synthetic procedures are usefull for readers. The presented manuscript should be accepted after extensive editing of the language. This Manuscript is well written and needs to minor revision.

Response:

Thank the reviewer for the good summary and valuable comments. We seriously make revisions based on your revision opinions point by point. We sincerely hope that our work can be published in the future after careful revisions. Individual response to each question is answered below.

Comment 1:

Every section of the manuscript must be written scientifically according to the published literature with appropriate references.

Response 1:

Thank you for your helpful suggestions. According to the published literature with appropriate references, we revised our manuscript scientifically. For instance, some changes as follow: We read every section of the manuscript and revised it carefully. We rewrite the first half of Introdction section to fully introduce the process of the experimental DDI validation and several strategies for predicting potential DDIs. We added some explanatory sentences into Methods section to improve the readability of our manuscript. We analyzed the potential mechanism of predicted DDI. We deleted some unnecessary references like ‘DrugBank 3.0: a comprehensive resource for ‘omics’ research on drugs’, ‘DrugBank 4.0: shedding new light on drug metabolism’, and ‘DrugBank: a comprehensive resource for in silico drug discovery and exploration’, etc.

Comment 2:

The logical flow of this manuscript is not perfect. The authors have written several matters haphazardly. The work appears as groundwork. Spacing, punctuation marks, grammar, and spelling errors should be reviewed wholly.

Response 2:

We appreciate very much the reviewer’s comments. We have revised the whole manuscript carefully and tried to avoid any spacing, punctuation marks, grammar, or spelling errors. In addition, we have asked several colleagues who are skilled authors of English language papers to check the English. We believe that the language is now acceptable for the review process. We also added many connecting sentences to make the manuscript more logical. Like the sentences of ‘Detailed description has been given below, according the order of function module in flowchart.’

Comment 3:

The flow of the introduction is not complete and unspecific. My recommendation is to construct the sentences more lucid and legible for more productive comprehension.

Response 3:

Thank you for pointing out this problem in the manuscript and giving valuable comments. To make the flow of the introduction more complete and specific, we rewrote some complex sentences that are difficult to understand to improve the readability of our manuscript. We highlighted the practical use of our model and added description information, which can make the readers easily understand our manuscript. Like the sentences of ‘In this chapter, the task of extracting embedding vector containing node structural in-formation is focused on. The node structure is more similar, the embedding vectors of the nodes are closer to each other’.

Comment 4:

The first paragraph of the introduction section contains no new information. Need to change.

Response 4:

Thank you for pointing out this problem in the manuscript. We have rewritten the Introduction section. We first descript the importance of predicting DDIs from the drug development perspective and clinical perspective. Then we elaborate on the process of the experimental DDI validation and several strategies for predicting potential DDIs. And we also demonstrate the disadvantages of experiments based on pharmacokinetics and the necessity of utilizing computational methods as a prescreening tool. Finally, we summarize existing computational methods and propose our model.

Comment 5:

sAuthor studied how to exploit network topology structure information and biochemical information to predict potential DDIs. On the one hand, case studies of three cancer-related drugs can indicate the good prediction ability of our model. On the other hand, our model could be seen as a pre-screening tool for potential DDIs. In this way, the work load of exploring unknown complex interactions of drugs can be reduced. In the future, to improve our framework, choosing negative samples with a more reasonable way to reduce the noise brought by unbalancing the original dataset and transferring our framework to predict interactions between unknown drugs would be considered. Is database screening possible for this model?

Response 5:

Thank you for your good summary. As the reviewer's consideration, our model can be used as a prescreening tool. For the researchers to further verify possible interactions in the perspective of biomedicine and pharmacology, we have trained the model with known DDIs and constructed a webserver, freely available at http://120.77.11.78/BioChemDDI/. In our web server, although we did not provide an interface for the database, it can perform the screening function for DDIs. In detail, when the reachers want to use our web server to perform preliminary screening, they should provide drug indexes and SMILES and submit them. If the users provided DrugBank ID for each drug, the calculation time will be shorter. The example can be seen on our webserver. Finally, the webserver returns the prediction DDIs of the top 100 score likelihood, and the results can be sent to the mailbox, provided by the users. The detailed tutorial also can be found in our webserver.

Round 2

Reviewer 1 Report

Authors sufficiently addressed reviewer's questions and comments. Even though the current manuscript is still composed with pure predictions, the reviewer agree that it tells a fair story about computational assessment of DDI. Experimental validation can be hold for future studies.

Author Response

Reviewer: 1

Comments and Suggestions for Authors:

Authors sufficiently addressed reviewer's questions and comments. Even though the current manuscript is still composed with pure predictions, the reviewer agree that it tells a fair story about computational assessment of DDI. Experimental validation can be hold for future studies.

Response:

The authors greatly appreciate your efforts and suggestions. Thank you for your recognition of our work and valuable comments. Just as the reviewer said, our manuscript tells a fair story about computational assessment of DDI. As a computer-aided drug discovery tool, it will help to face the challenges of the high cost, the limited participant number, the low efficiency and the large number of pairwise drugs waiting for identification. Moreover, in future work, we will fully conduct biological experiments to study DDI and validate the predicted DDIs. Additionally, we further reread and revised our sentences to ensure the readability of our manuscript.

Reviewer 3 Report

The revised version of the manuscript includes all remarks and modifications indicated. The main concerns of the manuscript have been solved. In my opinion, the provided version is now suitable for publication

Author Response

Reviewer: 3

Comments and Suggestions for Authors:

The revised version of the manuscript includes all remarks and modifications indicated. The main concerns of the manuscript have been solved. In my opinion, the provided version is now suitable for publication

Response:

The authors greatly appreciate your efforts and suggestions. Thank you for your recognition of our work and valuable comments. In future work, we will further improve our framework and do more studies on the prediction of potential DDIs. Additionally, we also further reread and revised our sentences to ensure the readability of our manuscript.
